# Reading in COVID-19 Pandemic Times: A Snapshot of Reading Fluency of Portuguese Elementary School Students

**DOI:** 10.3390/children10010143

**Published:** 2023-01-11

**Authors:** Daniela Rosendo, Armanda Pereira, Tânia Moreira, José Carlos Núñez, Joana Martins, Sílvia Fróis, Conceição Paupério, Pedro Rosário

**Affiliations:** 1Department of Applied Psychology, Psychology Research Center, School of Psychology, University of Minho, 4710-057 Braga, Portugal; 2Department of Education and Psychology, School of Human and Social Sciences, University of Trás-os-Montes and Alto Douro, 5000-801 Vila Real, Portugal; 3Faculty of Psychology, University of Oviedo, 33003 Oviedo, Spain; 4School Psychology Office, Agrupamento de Escolas Sá da Bandeira, 2005-191 Santarém, Portugal; 5Elementary School Department, Agrupamento de Escolas de Campo, Valongo, 4440-201 Porto, Portugal

**Keywords:** reading, reading fluency, setbacks, learning, pandemic times, summer loss

## Abstract

The development of reading skills foresees fluency in reading. Prior research has shown that during periods of absence from school, students are prone to showing setbacks in their learning. However, the literature presents mixed findings, possibly explained by the families’ socioeconomic statuses. The present study aims to analyze fluctuations in learning acquisition, specifically in reading fluency, during the pandemic, when all students were absent from school for several months. Data were collected in two waves. The present study combines quantitative and qualitative data with an explanatory sequential approach. Participants were 52 3rd-graders and their teachers. The latter were enrolled in two-member check sessions. Statistically significant differences in speed (lower than expected) and prosody (higher than expected) were found between the two sessions. Considering ASE support (financial support for low-income families from the Portuguese government), data indicate that students benefiting from this support showed performances in accuracy and speed below the expected. Prosody scores were above the expected at both sessions. Findings may provide relevant insights to further understand fluctuations in students’ reading fluency during long periods of absence from school; for example, data could help prevent learning setbacks due to summer vacations.

## 1. Introduction

Learning to read is one of the main goals in the early years of elementary school. Reading ability is considered a rooting learning skill, which is essential for a wide range of daily life activities; for example, following subtitles in a movie, reading, and understanding a letter from the tax office and is, therefore, closely related to learning in the school context [1,2].

Reading fluency is permeable to instruction and training, which means that under these conditions, students are likely to improve reading fluency [3]. Fluent readers master three main elements of reading fluency and typically read with appropriate accuracy, speed, and prosody [4]. Reading speed is determined by the number of words read per minute throughout the reading session [5,6]. Reading accuracy refers to the reader’s ability to decode orthographic forms and produce words accurately [6]. Finally, reading prosody refers to the reader’s ability to read orally with appropriate expression or intonation to maintain the meaning of sentences [7,8]. To further understand individuals’ reading competencies, prior research has focused their investigation on the role played by reading literacy [9,10]. Reading literacy is the ability to use, understand, evaluate, reflect on, or engage with texts to achieve goals or develop knowledge about a particular topic [2]. Individuals need to read proficiently to develop their reading literacy ability [11,12]. For example, Portuguese data from 2018 show that 21% of 15-year-old students did not reach the basic level of reading literacy [2]. Moreover, seven percent reached a high level of proficiency, and only 0.8% surpassed the maximum level of proficiency [2]. 

Recent OECD reports [12] warn that more than reaching proficiency in reading fluency seems to be needed. Maintaining reading fluency requires systematicity and involvement in reading practice. Moreover, children who do not engage in reading-related activities systematically may show problems in their reading skills development. Therefore, instigating reading habits prevents this skill from being downgraded [13]. For example, during an eight-week summer intervention, Pagan and Sénéchal [14] encouraged children from the third and fifth grades with low reading comprehension skills and poor vocabulary to read one book per week.

Additionally, parents of these children were trained to promote reading comprehension at home. Data indicate enhancements in reading comprehension, fluency, and vocabulary acquisition [14]. Prior studies emphasize that the amount of time and frequency of activities involving reading are related to improvements in reading performance [13,15]. These practices can be particularly relevant during out-of-school periods; for example, during the school holidays, when a decrease in the children’s literacy skills is likely to affect their reading fluency [13,16].

During periods of absence from school, schools and teachers typically do not reach children and, consequently, play a minimal role in their education during these time frames [17]. The literature and ad hoc evidence indicate that these periods can negatively impact prior learning achievements (i.e., students return to school with less content domain knowledge than when they left) [18]. Interestingly, Angrist and colleagues [19] advanced this discussion by stating that this impact also corresponds to a cost in opportunities for learning (i.e., the amount of learning that students would have acquired during a typical school year). 

Summer vacations (in Portugal, about three months away from school), public health crises (e.g., pandemic periods), and natural disasters (e.g., hurricanes) represent three scenarios prone to instigate negative impacts on acquired learning [17,20]. Atteberry and McEachin [17] suggested that, during summer, more than half of the students from first to eighth grade are likely to show learning setbacks. Learning setbacks translate to a slowdown or downturn in academic learning due to the summer period [21]. Moreover, Vale et al. [22] set a longitudinal study in Australia with a sample of approximately 2500 children between the third and seventh grades to investigate their reading achievement in the holiday months. Results indicated a slowdown in these children’s reading achievement over the summer holidays.

There is extensive solid research on learning setbacks during summer holidays [23]. One explanation for summer learning setbacks is the disparity in the availability of academic materials over the summer. Importantly, families from low socioeconomic status (SES) backgrounds have fewer opportunities to access educational materials over the summer than their counterparts [23,24]. von Drehle [25] reported that for students from high socioeconomic (SES) backgrounds, the summer holidays provide opportunities to expand their learning experiences (e.g., traveling in foreign countries, practicing foreign languages, enrolling in educational summer programs). Contrarily, students from socio-economically disadvantaged backgrounds do not have similar opportunities. Consequently, by the end of elementary school, students from low-income backgrounds are likely to be nearly three grade levels behind their counterparts [25].

Hurricane seasons and public health crises close schools for several weeks or months and are likely to impact students’ learning losses, similar to summer holidays. These periods negatively impact learning, increasing the probability of students dropping out of school [20,26,27]. For example, during the 2005 Katrina and Rita hurricanes, evacuated children had significant impacts on their academic performance and school trajectories. Due to these natural catastrophes, students missed, on average, about five weeks of school [28]. Similarly, in 2009 during the H1N1 pandemic health crises in São Paulo, the school closures led to the extension of winter holidays, which negatively impacted student development of academic-related competencies (e.g., Portuguese language and mathematics proficiency, Amorim et al. [20]). Recently, we experienced a period with an impact on the educational systems worldwide with short- and long-term implications. Some evidence about the negative effect of potential learning setbacks during this period of at-distance school is already beginning to emerge. For example, a study in Norway indicated that first graders during the COVID-19 pandemic showed lower scores on writing quality, handwriting fluency, and attitude toward writing than their counterparts who completed the 1st grade in the previous year [29]. Additionally, during the COVID-19 pandemic, Klosky and colleagues (2022) explored in Georgia the effects of distance education on 3rd-grader’s learning (i.e., at this stage, reading proficiency is shown to be a significant predictor of future academic achievement). Results showed that most distance education barriers were technology related (e.g., internet problems, difficulties in focusing attention, Zoom fatigue). These difficulties ended when children returned to face-to-face school; parents reported improvements in their children’s academic performance [30].

During the school closure period due to the COVID-19 pandemic, students spent less time interacting with teachers and more time following school at-distance. Some were accompanied by their parents, while others received limited family support [31]. At-distance schools emphasized the families’ socioeconomic inequalities and contributed to diminishing the learning opportunities for students from less resourceful backgrounds. This social scenario is particularly relevant because the UN reports predicted an increase in poverty globally due to the COVID-19 pandemic [32].

In Portugal, since March 2020, school support for students (e.g., dinner meals, meals delivered on the weekends, internet kits) has increased by about 50% [33]. The reasons behind these data may be various. For example, families limited material resources (e.g., many families lost their jobs during this period, limited internet connection at home). As parents and teachers report, at-distance school experience negatively impacted students’ learning process.

Despite this worrying educational scenario, a detailed analysis reveals that some skills seem particularly prone to be undermined during a time out of school, namely, skills requiring systematic practice sessions, such as reading competence [6,23,34]. Previous research shows that one of the short-term impacts of at-distance school is the resulting learning setbacks, which can lead to long-term impairments in reading and writing skills, particularly in students with a history of poor achievement [35]. This ballast may compromise children’s aspirations for their future [36]. This investigation occurred just after the 2020 summer holidays amid the COVID-19 pandemic context. Quantitative and qualitative data were combined in an explanatory sequential approach. This study analyzed the fluctuations in 3rd-grade children’s acquisition, specifically those focused on reading fluency. 

Why 3rd-grade students? At this stage, reading becomes a way to convey the content of academic subjects rather than a way to practice that skill [37]. Why focus on reading fluency analysis? As aforementioned, reading fluency is closely related to instruction and practice, and during the COVID-19 pandemic period, these conditions were not accessible to all students [38,39]. Few studies focused on collecting and analyzing the fluctuations in these core educational skills and, consequently, on the factors that led to lower performances of this skill. Current findings are expected to add to the literature on learning setbacks in reading fluency during long periods of absence from school.

This study’s research questions are threefold: (i) How did 3rd-grade students perform in reading fluency after the 4-month lockdown (at-distance school) compared to what is expected for reading fluency at this grade level?; (ii) Are there differences in students’ reading fluency performance related to the ASE [Ação Social Escolar] support (see definition in the Section 2.1 of this manuscript)?, (iii) Are there differences in students’ reading fluency three months after returning to in-presence schooling?

To answer our research questions, two different sessions were considered: after the first lockdown (June—[M]oment 1), and three months after the return to in-presence schooling (November—M2).

Hopefully, this study’s results will contribute to deepening our understanding of the extent of the consequences of out-of-school time during the COVID-19 pandemic for the fluence reading trajectories of Portuguese elementary school students. We believe the results will add to the previous literature and contribute to creating guidelines to help practitioners and inform educational intervention tools to mitigate long-term negative consequences in this generation of students.

## 2. Materials and Methods

An explanatory sequential approach was conducted to assess the impact of absence from school on students’ reading abilities.

### 2.1. Participants and Setting

Four elementary public schools in a northern region of Portugal and five in a southern central area region of Portugal were invited to participate in the study; all agreed to participate. Finally, this study included 12 3rd-grade classes. Each class was composed of approximately 22 students. From each class, five to seven students were randomly selected to enroll in the study. A total of 52 participants (26 males and 26 females) aged between eight and 10 years old (*M* = 8.56; *SD* = 0.53) were included in the sample. From this sample, 14 had ASE support. ASE support is a financial allowance (e.g., free school meals or materials) provided by the Portuguese government to help families from low SES contexts. The main objective is to combat school dropout and promote equal opportunities for individuals to access education. The eligibility criteria to benefit from ASE support are related to family income [40]. All the participants were native speakers of European Portuguese. The teachers of these children were enrolled in two focus groups (see Section 2.3.3), with six participants each. Participants were all female, aged between 39–62 years old, and held an average of 24.5 years of teaching experience.

The current study was conducted during the COVID-19 pandemic between June and November 2020 (Note: in Portugal, at-distance learning took place from March to July 2020, students attended in-person classes from September 2020 onwards). The World Health Organization (WHO) classified, in March 2020, COVID-19 as a pandemic [41]. COVID-19 spread quickly around the world, requiring strict public health measures to mitigate the spread of the disease [42]. One of the most significant measures worldwide was the closure of schools [43], which was profoundly disruptive to the educational systems [44].

On 16 March 2020, the Portuguese government decided to close all schools in the country. This measure led to the need to instigate a rapid and sudden adaptation to an unprecedented emergency educational response—distance learning, for which no school or teacher was prepared. To cope with the sudden transition from in-person to distance learning, several initiatives were developed in Portugal (e.g., ‘Roadmap—Guiding Principles for the Implementation of Distance Learning E@D in Schools’, #EstudoEmCasa (Study at Home), Yellow Trials and Tribulations, COVID-19 in Trials and Tribulations) [45,46,47]. These educational initiatives took place from March to July 2020.

### 2.2. Explanatory Sequential Approach

The present study followed an explanatory sequential approach. Our purpose was to use qualitative data after the quantitative data collection to expand our understanding of how this specific period (school closure due to the pandemic) impacted the students’ reading abilities. Quantitative (QUAN) data collection and analysis were conducted before the qualitative (QUAL) data collection and analysis. In the qualitative phase, a focus group was run to “unpack” the quantitative results regarding teachers’ perspectives and field experience. Importantly, the findings of both phases were used to reach conclusions.

### 2.3. Data Collection and Procedures

A launch meeting with the teachers of the classes included in the study was set before data collection. This meeting aimed to clarify the study’s goal and explain the data collection procedures. Afterward, teachers who agreed to participate in the study informed parents/guardians of their students about the research and asked them to sign an informed consent form allowing children to participate.

Two data collection waves were set with students in the aftermath of parents/guardians signing informed consent forms. Teachers collected data in their classes. These teachers received training on this particular data collection protocol (e.g., Text Reading Fluency and Prosody) from the research team previously to data collection sessions. The first session occurred in June 2020, at the end of the 3rd-grade and just before the summer vacation; the second session occurred in November 2020, in the middle of the first term of the 4th-grade. After these data collection sessions with students, the teachers’ online focus group occurred in July 2021. Two online focus groups were conducted, one for each school grouping. 

The assessment protocol for quantitative data collection included: text reading fluency (accuracy and speed), prosody, and reading academic target-skills assessment. For the qualitative data collection (focus groups), a semi-structured script was designed to deepen the comprehension of the quantitative results. 

#### 2.3.1. Quantitative Data Sources

Text reading fluency. Three texts were selected from one of the national schoolbooks used for 3rd-grade classes in public schools. Participants did not know these texts. 

All texts were written in Times New Roman 14 font type, with standard spacing between letters and 1.5 spacing between lines. Each text was analyzed through a readability formula. The readability level of each text was assessed by the Flesch Reading Ease Formula, adapted for the Spanish language [48]. There is no readability formula available for the European or Brazilian Portuguese language; the adapted formula for the Spanish language was used because it shows a high degree of similarity with the Portuguese language regarding the frequency of mono and multisyllabic words [48,49]. The adapted Flesch Reading Ease Formula [50] considered for the readability calculation was the following:*R* = 206.84 − 0.60 *ASW* − 1.2 *ASL*

In this adapted Flesch Reading Ease Formula, corrected by Law in 2011, the *R* stands for the readability value, *ASW* stands for the average number of syllables per word, and *ASL* for the average sentence length. The higher the *R* score, the easier the text to read. Scores range from 0 (very difficult to read) to 100 (very easy to read).

The length of the three texts selected for the current study was 216, 339, and 348 words, respectively. Text 1 had an R score of 83.1 (easy), text 2 of 84.56 (easy), and text 3 of 85.19 (easy).

In addition to the R-value, the correspondence between the text and the school level was calculated through the Crawford formula [51]. Finally, data showed that the readability of the three texts was fit to this sample, 3rd/4th-grade level.

Prosody. To assess the four dimensions of prosody: expression and volume, phrasing, smoothness, and pace, we used an adapted version of the Multidimensional Fluency Scale with a Cronbach alpha of 0.96 [52,53]. Teachers evaluated each student’s oral reading performance level for each dimension in the three texts. The Multidimensional Fluency Scale [45] provides four levels: level one stands for poor, and level four for high competencies. For example, in the pace dimension: level one stands for “slow and laborious”; level two for “moderately slow”; level three for “uneven mixture of fast and slow reading”, and level four for “consistently conversational” performance. Scores on this scale range from four to 16, with a cut-off of eight. To ensure teachers assessed students’ prosody following a similar protocol, all participating teachers received training on using this instrument from the research team before this study. During this training, teachers independently coded the oral reading of students not enrolled in the study and discussed disagreements until reaching a consensus. Overall, teachers working in pairs reached an inter-rater agreement ranging from 0.80 to 0.87.

Reading Academic Target-Skills. Teachers assessed each 3rd-grade student’s reading performance using a curriculum-based checklist composed of four items (e.g., “Read all regular monosyllabic, disyllabic, and trisyllabic words and, with few exceptions, all irregular words found in texts used in school” or “Read a text with correct articulation and intonation and a reading speed of at least 110 words per minute”). Teachers rated each item by providing quality ratings (e.g., “far below grade-level expectation” and “far above grade-level expectation”). This checklist follows the Portuguese Ministry of Education reading goals for 3rd-grade students [4,54].

Procedure. For the assessment of the two dimensions of reading fluency (accuracy and speed), teachers asked students to read each text for 60 s. The reading was audio-recorded. During the first data collection moment, June 2020, audio-recorded readings were done through Zoom^®^, Skype^®^, Teams^®^, or other applications used during the at-distance learning. In the second data collection moment, the audio-recorded readings were run in person, at school, and recorded through cell phone/computer applications. After the audio-record readings, teachers were invited to fulfill an assessment protocol per student, including reading fluency, accuracy, and prosody data. For the assessment of accuracy and speed, we took into consideration the following errors: (i) Omission—when the child did not read some word(s); (ii) Mispronunciation; (iii) Substitution—when there is a change in words or sentences, and (iv) Letter swapping—when students changed letters of position and (v) the number of words read by the teacher after 2–3 s without an adequate response.

Two researchers listened to the audio-recorded students’ reading and independently assessed the number of words read and errors to enhance the trustworthiness of the reading fluency and accuracy assessments. Finally, they reached an inter-rater agreement of 0.79, considered a strong agreement [55].

Additionally, teachers were asked to provide students’ socio-demographic information (age, sex, grades in the previous in-person Portuguese language test, and ASE support). Teachers were also asked to report the perceived quality of students’ reading according to the curricular goals set by the Portuguese ministry of education. All data were collected through a Google Forms^®^ survey.

#### 2.3.2. Quantitative Data Analysis

Data analysis was carried out with different tests regarding the properties of the variables and the type of analysis required to address the hypotheses. Initially, we examined the distribution of the variables. Subsequently, to learn whether non-benefiting or benefiting from ASE support had statistically significant effects on the dependent variables or whether the scores of each group were statistically different from the expectations for this education level, we used a T-Test (for a sample or independent samples). Moreover, and following Rasinski [52], to analyze the accuracy, we considered three levels of performance, i.e., the percentage of words read correctly per minute: (i) independent level (97 to 100%); (ii) instructional level (90 to 96%), and (iii) frustration level (<90%). To analyze speed, we considered the number of words read per minute. Third-grade students are expected to read about 110 words per minute [56].

To analyze whether the change in the levels of the dependent variable between the two measures (June and November 2020) was similar for both groups of students (non-benefits or benefits of ASE support), we used a multivariate analysis of covariance (MANCOVA), taking the levels of the dependent variables measured in June as covariates. The effect size was assessed according to the criteria established by Cohen [57] (small, *d* = 0.20; medium, *d* = 0.50; and large, *d* = 0.80), and the partial eta squared in the results of the multivariate were analyzed as follows: small = 0.010; medium = 0.059; large = 0.138).

#### 2.3.3. Qualitative Data Sources

An online focus group with teachers was conducted to contextualize the quantitative findings and deepen the interpretation and comprehension of the results.

Procedure. After participants signed informed consent, the significant and non-significant quantitative results were presented to teachers who were asked about their perceptions of the meaning of the results and whether these results represented their understanding of students’ reading behavior in class. Prompts were, for example: “Do you believe these results represent your perception of students’ reading skills in this period? Can you elaborate?”, “How can you explain these results?”, and “Which are the implications to the educational practices?”.

Focus groups were conducted by the first author, who was helped by two researchers taking notes during the sessions. Trained research assistants transcribed the two focus groups verbatim.

#### 2.3.4. Qualitative Data Analysis

The data analysis followed a thematic approach to identify common themes across participants’ discourse [58]. Two researchers read the two focus group transcriptions independently to register their first thoughts about the data. The data was read a second time to identify and list significant statements. The next step was identifying statements and labeling them into units or themes. Researchers discussed all discrepancies until they reached an agreement. An external auditor informed about the study goals was invited to analyze the themes in which the agreement was not reached. Afterward, to increase the reliability of the current findings [59], two researchers coded the complete content of one of the two focus groups separately and reached a consensus through discussion. The consistency of coding was assessed by Cohen’s kappa coefficient. The kappa coefficient was 0.92, which is considered almost perfect, according to Landis and Koch [55]. Verbatim quotes were included to illustrate the data captured, support discussion points, and add validity to the results. To clarify the description of the categories regarding the frequency of responses, Rodgers and Cooper’s [60] scoring scheme was used as follows: ‘All’ = 100%; 12 cases, ‘nearly all’ = 100% − 2; 10 cases, ‘most’ = 50% + 1 to 100% − 2; 7 to 10 cases, ‘around half’ = 50% + 1; 7 cases, ‘some’ = 3 to 50% + 1; 3 to 7 cases, ‘a couple’ = 2 cases, and ‘one’ = 1 case. QSR NVivo 10 software was used to help with the coding and data analysis [61].

## 3. Results

### 3.1. Quantitative Results (QUANT)

#### 3.1.1. Preliminary Analysis

Table 1 provides descriptive statistics corresponding to the variables analyzed, i.e., accuracy, speed, prosody, perceived quality, and Portuguese language subject achievement.

#### 3.1.2. *T*-Test Analysis

*T*-Test analysis for independent samples was conducted to answer two questions: Are the accuracy, speed, and prosody levels after the lockdown (June) significantly lower than they typically would be before the pandemic situation (setbacks assessment)? Is the level of learning setbacks similar for students with and without ASE support? 

The results obtained for the total sample indicated no statistically significant differences between the current and normative scores in accuracy, neither in June, M1 (DiffM [differences between means] = −0.913; t = −0.673; *p* > 0.05), nor in November, M2 (DiffM = 2.261; t = 1.748; *p* > 0.05). However, statistically significant differences were found for the speed variable, both in June (DiffM = −20.801; t = −5.906; *p* < 0.001) and November (DiffM = −2.081; t = −7.935; *p* < 0.05), lower than expected in both cases. Note that 3rd-grade students are expected to read about 110 words per minute. As for the prosody variable, statistically significant differences were also observed at both time points; in both cases, the level is higher than usual: June (DiffM = 0.707; t = 7.459; *p* < 0.001), November (DiffM = 1.050; t = 8.934; *p* < 0.001).

Next, we analyzed data considering participants’ ASE support; results complement those previously reported. 

Accuracy—Individuals with ASE support showed a lower accuracy than expected, both in June (DiffM = −9.958; t = −3.827; *p* < 0.001) and November (DiffM = −6.409; t = −2.407; *p* < 0.05). Contrarily, participants without ASE support level of accuracy were significantly higher than expected in June (DiffM = 3.875; t = 5.493; *p* < 0.001) and November (DiffM = 6.850; t = 16.299; *p* < 0.001). The differences found between June and November were statistically significant in both conditions, i.e., the ASE support group (DiffM = −3.549; t = −2.255; *p* < 0.05) and the without ASE support group (DiffM = −2.975; t = −3.872; *p* < 0.001); note that data for accuracy were higher in November for both groups.

Speed—Results indicated that participants with and without ASE support reading fluency were below what would be expected at the first evaluation (June); (DiffM = −31.944; t = −6.760; *p* < 0.001) and (DiffM = −14.901; t = −3.317; *p* < 0.01), respectively. At M2, participants with ASE support showed data below the mean (DiffM = −20.018; t = −3.972; *p* < 0.01), and participants without ASE support showed results near to the mean (DiffM = −1.539; t = −0.315; *p* > 0.05). In both groups, a statistically significant increase was observed between June and November: with ASE support (DiffM = −11.925; t = −4.522; *p* < 0.001); without ASE support (DiffM = −13.362; t = −5.917; *p* < 0.001).

Prosody—Both individuals with and without ASE support performed significantly above the mean in June (with ASE support: DiffM = −0.430; t = 3.069; *p* < 0.01; without ASE support: DiffM = 0.853; t = 7.215; *p* < 0.001) and November (with ASE support: DiffM = 0.608; t = 3.187; *p* < 0.01; without ASE support: DiffM = 1.284; t = 9.590; *p* < 0.001). Participants without ASE support showed an increase in their prosody between June and November (DiffM = −0.430; t = −4.417; *p* < 0.001), while participants with ASE support only showed a tendency to increase their scores (DiffM = −0.177; t = −1.838; *p* > 0.05). 

#### 3.1.3. Multivariate Analysis of Covariance (MANCOVA) Analysis

The following questions: (i) are there statistically significant differences between participants with and without ASE support in their accuracy, speed, prosody, performance, and perceived quality in June and November? (ii) is the trend of the DV (Dependent Variables) accuracy, speed, prosody, performance, and perceived quality similar between June and November? were answered through a multivariate analysis of covariance (MANCOVA).

The MANCOVA analysis was run using the DV levels in November, a measure of the DV’s, and as covariates, the measures of these variables in June. The aim was to determine whether the measures of both groups (with and without ASE support) in November differed significantly after controlling statistically the differences observed in June. At the multivariate level (considering the five dependent variables together), results indicated that there are no statistically significant differences between the two groups of participants (Wilks’ Lambda = 0.807; F(44) = 1.910; *p* > 0.05; eta squared partial = 0.193). From a univariate perspective, it was observed that there were no statistically significant differences between the two groups in accuracy (F (44) = 0.365; *p* > 0.05; eta squared partial = 0. 008), speed (F (44) = 0.231; *p* > 0.05; eta squared partial = 0.005), prosody (F (44) = 0.034; *p* > 0.05; eta squared partial = 0.001) and performance F (44) = 0.196; *p* > 0.05; eta squared partial = 0.004). However, statistically significant differences in perceived quality were found (F (44) = 7.163; *p* < 0.01; eta squared partial = 0.140), with a large effect size. Figure 1a–e shows the parallelism of the slopes.

### 3.2. Focus Group Results (QUAL)

The decision to resort to focus group as a data collection method was grounded in three reasons: (i) return quantitative findings to teachers, (ii) generate new data through the teacher’s interpretation of the quantitative data, and therefore, (iii) extend the quantitative findings (QUAN).

Data were analyzed and related to the research questions to find themes. Two themes were found while analyzing teachers’ perspectives on the shared results of the quantitative study.

Theme 1: Away from the school, away from the books

Regarding the first two research questions, i.e., analysis of the student’s performance against curriculum goals while considering the *ASE* support, the quantitative results revealed that without differentiating the ASE support received by the families of the students, differences were found in speed (lower than expected) and prosody (higher than expected). Considering the ASE support received, those without ASE support showed a performance in accuracy and speed below average, but prosody scores remained above average. All teachers agreed with these results and mentioned that they resembled their experience with the students during this period. Reasons that might explain findings were mainly focused on student-related variables. There seemed to be a consensual agreement on aspects directly linked to reading (e.g., interest/pleasure for reading, expressiveness in reading, time devoted to reading),

B2FG1: “*there are very few students who enjoy reading for pleasure. [For most] It is an obligation*”.

However, also on general aspects (e.g., distraction, previous learning difficulties, type of vocabulary, student maturity, motivation) that negatively impact students’ reading fluency performance.

B1FG1: “*… when at a distance they do these games, you know, changing screens, being distracted…,* [while having classes at home] *there was more distraction. There were more distractors…*”.

Most participant teachers attributed students’ positive conquests in their reading fluency performance to their efforts (i.e., teacher involvement). For example, efforts were made to reinforce reading at school and home, teacher-guided reading, and exploration of integral works at school. In addition, teachers stressed efforts made during the at-distance learning period to provide direct support to students or their parents while delivering strategies to help them cope with their children’s needs.

B7FG2: “*… we as teachers provide them strategies, we teach them how they can stimulate children to read, what they can do to promote reading at home…*”.

About half of the participants also attributed the positive results to school resources (e.g., school projects, visiting the library and borrowing books, and providing computers during lockdowns). Participants attributed the less favorable results to contextual factors (e.g., more distracting factors in distance learning, vocabulary specific to the generation, pandemic health conditions, and family’s reading culture).

B4FG1: “*With the return to the in-person school, it was also possible to return, for example, to the library visit sessions … many book requests from children…*”.

B7FG2: “*During the school year, we had an unstable presence at school*”.

Despite participants’ consensual opinions on the results presented previously, some participants did not associate the lower levels of performance in reading accuracy and fluency of students with ASE support. Instead, they were more prone to associate these differences with family related factors, such as family time management, parents’ expectations, and families’ educational priorities (e.g., providing smartphone gaming apps instead of books). Some teachers also underlined that during the at-distance learning period, the schools supported students from families with ASE support in several dimensions (e.g., provision of computers, printed study materials, and dinner meals). 

B6FG1: “*(…) with the help of the municipality, the school provided the computers to children to help them follow the classes… but we also had parents that helped other parents to create emails, to access the platform, to teach children through their cell phones…*”.

Theme 2: Reading trajectories in pandemic times

Upon presentation of the quantitative results concerning the second research question (i.e., associated with student performance between M1 and M2 and considering the ASE support), most participants recognized these results in their educational practice. The quantitative results revealed no differences between the two groups (with and without ASE support) in accuracy, speed, prosody, or performance. Most participants reported that findings were a good picture of their students’ reading behaviors; however, some participants refrained from answering.

All participants who agreed with the results underlined their involvement, particularly learning how to read efforts and opportunities to improve reading skills, as key to the improvement of their students’ learning practices. These teachers also mentioned that despite their efforts and dedication to instigating good practices and improving students’ reading performance, sanitary constraints during the pandemic impacted students’ performance. “*This instability that we lived in, haaa… and the conditions in which we were working (...) created instability, and all considered it was very difficult to focus on learning.*” (B7FG2). All teachers acknowledged that returning to in-person learning facilitated learning activities and student development. These teachers also mentioned that after returning to in-person learning, they needed to direct their efforts toward students with more difficulties. “*I think we invested much time in this more fragile group, with those with more learning weaknesses. This was notorious.*” (B4FG1). Teachers mentioned that this particular aspect may have contributed to the fact that students with more difficulties showed an increase in their performance after returning to in-person learning activities.

B4FG1: “*But, maybe, haaa haaa, the others* [students with higher grades] *may have felt a little bit resentful of that* [paying more attention to students struggling to read and learn]*, also the boys with more capacity… it is possible that we have left them a little bit uncovered. The investment in the most fragile group was very strong*”.

Some teachers reported that students showing good grades who received limited support had the resources to improve their performance autonomously and were among the more motivated to return to in-person classes. “*The truth is, the more capable ones, sooner or later, they succeed; little by little, they get to the level of mastery where they always have been*.” (B4FG1). In addition, returning to in-presence classes after the lockdown and summer vacations allowed these students to access school resources likely to help them enhance learning. “*(...) from September onwards, we had this idea to work very closely with the school library*.” (B1FG1).

## 4. Discussion

The worldwide COVID-19 pandemic has impacted the educational systems with short- and long-term implications. During this critical period, students attended at-distance classes, negatively impacting learning outcomes. For example, evidence has already been gathered showing the negative impact of potential learning setbacks during the at-distance learning window [29]. One of the possible impacts of distance learning, particularly for lower-achieving students, is learning setbacks [32]. Importantly, learning setbacks may lead to severe difficulties in learning in the long term. All things considered, it became important to deepen our understanding of the fluctuations in learning acquisitions after the four months of at-distance learning and some months after the return to in-presence school due to the COVID-19 pandemic context and the extent of the consequences of this unique scenario. Acknowledging the challenges of this unique scenario and considering that reading is a skill that can be severely affected when students are out of school (i.e., due to the need for constant practice and feedback), the current study is expected to provide educators with important insights on students reading fluency during long periods of absence from school.

Current findings showed no statistically significant differences in M1 and M2 between current and expected performance in accuracy. However, differences were found in speed (lower than expected) and prosody (higher than expected). The latter might be explained by the fact that teachers assessed their student’s prosody, and despite following the Multidimensional Fluency Scale [52], their assessments could have benefited students. This research intended to prevent this possibility by training teachers to code this task rigorously; still, teachers could have scored students higher than their actual performance. Data collection, particularly the first wave, was gathered during a long period of at-distance school; for this reason, teachers were the educators with more proximity to students and their families, and thus were those more prone to deliver this research. 

Analyses differentiating the ASE support provided have shown that (i) those students with ASE support performed significantly below average in accuracy (M1 and M2, although significantly lower in M1 than in M2); in speed (M1 and M2, lower in M1 than in M2), but above average in prosody (M1 and M2, although higher in M2, differences were marginal), and there were no differences in teacher-perceived quality between M1 and M2; (ii) those without ASE support yield significantly higher in accuracy (M1 and M2, although significantly more in M2), in fluency (below average in M1, but not in M2, these differences being statistically significant), but above average in prosody (in M1 and M2, higher in M2), and statistically significant differences in quality perceived by teachers between M1 and M2 (in favor of M1). The qualitative focus group results indicated that our study mainly focused on student-related variables (e.g., time to read). Consistent with previous studies, children who do not engage in reading activities are likely to lag in developing reading skills [13,15]. In addition, and consistent with prior studies, participants also attributed data to their involvement in the students’ learning (in-person or at-distance learning). When students are disconnected from the elements of the school, this disengagement can harm prior learning [17,18,19]. The quantitative data showed that when ASE support is considered, there are differences in accuracy and speed. This result could be due to the poor availability of materials to help students learn; and lack of practice [23,24] during the at-distance period.

Additionally, statistically significant differences were found in M1 and M2 between the two groups of participants in all the independent variables (in favor of those without ASE support). However, the differences observed in M2 were similar to those observed in M1, so it can be stated that the change occurring between M1 and M2 is similar in both groups. This trend, in general, showed a significant increase in the dependent variables from M1 to M2 in both groups. That is, the slopes are parallel (see Figure 1). The similarity of the trend in both groups between the two-time points is inconsistent with the previous literature. It would be expected that students with more available resources (without ASE support) would have a higher growth curve between M1 and M2. Participant teachers provided one possible explanation for this finding; after returning to in-person classes, teachers dedicated more time and effort to students with more needs and difficulties. 

In sum, our findings support the idea that, during periods in which students are away from school, the practice of reading needs to be instigated (e.g., promoting involvement in reading-related activities). For example, Pagan and Sénéchal [14] encouraged 3rd- and 4th-graders to read one book weekly. In addition, it also seems important to promote parental and family involvement. As previously suggested by Pagan and Sénéchal [14], encouraging parental involvement promotes children’s reading comprehension, fluency, and vocabulary acquisition.

Finally, despite the challenges presented by the unexpected closure of schools due to the COVID-19 pandemic, we observed promising results. In fact, despite all the obstacles, the Portuguese educational system was able to readjust and act according to the new COVID-19 plans. Current data on students show that receiving feedback and support (from parents and teachers) helped them acquire reading skills. Policymakers and legislators could consider providing extra support (e.g., delivering computers with internet free, hiring teachers) for needier families with low-literacy parents. Findings are expected to inform future research and social policies focused on improving distance education in future similar situations, promoting more positive and facilitative learning experiences for all social and economic statuses.

## Figures and Tables

**Figure 1 children-10-00143-f001:**
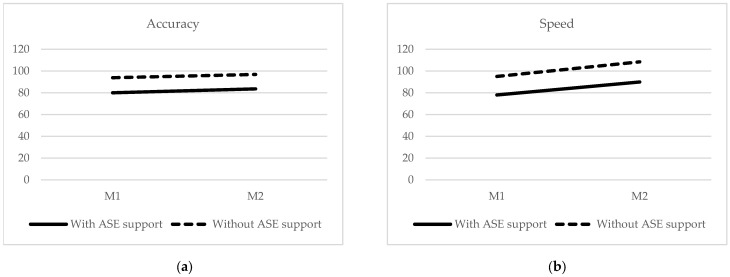
Graphics (**a**–**e**) show the differences between participants with and without ASE support between June and November. (**a**) Results for accuracy measured in June (M1) and in November (M2) for participants with and without ASE support. (**b**) Results for the speed measured in June (M1) and November (M2) for participants with and without ASE support. (**c**) Results for prosody measured in June (M1) and November (M2) for participants with and without ASE support. (**d**) Results for perceived quality measured in June (M1) and in November (M2) for participants with and without *ASE support*. (**e**) Results for performance measured in June (M1) and in November (M2) for participants with and without ASE support.

**Table 1 children-10-00143-t001:** Mean (M) and standard (SD) deviation of the variables under examination.

	With ASE Support		Without ASE Support		Sample Total	
	*M*	*SD*	*M*	*SD*	*M*	*SD*
Accuracy						
M1	80.04	11.04	93.87	4.11	89.08	9.78
M2	83.59	11.29	96.85	2.45	92.26	9.32
Speed						
M1	78.05	20.04	95.09	26.19	89.19	25.39
M2	89.98	21.38	108.46	28.48	102.06	27.50
Prosody						
M1	2.43	0.59	2.85	0.69	2.70	0.68
M2	2.60	0.81	3.28	0.78	3.05	0.85
Perceived quality						
M1	2.40	0.72	2.94	0.71	2.75	0.75
M2	2.46	0.61	3.28	0.57	2.99	0.70
Performance						
M1	66.72	13.85	84.06	10.54	78.06	14.32
M2	69.17	11.34	80.58	12.65	76.55	13.28

Note: M1 (June), M2 (November).

## Data Availability

The data presented in this study are available on request from the corresponding author. The data are not publicly available due to privacy and ethical reasons.

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
