# Peer review of "Reading in COVID-19 Pandemic Times: A Snapshot of Reading Fluency of Portuguese Elementary School Students"

_children, 2023, doi:10.3390/children10010143_

Round 1

Reviewer 1 Report

Dear Authors,

I am very pleased to review such an article. I think it touches on a crucial title.

However, as a non-Engllish speaker, I have many confusions and questions in my mind. In my opinion, some parts of the article should be improved.

1-In the abstract, you said 22 third graders included. In the methods section, you wrote 52. 

2-In the abstract, you mentioned accuracy and speed but skipped prosody and the other two variables. Why?

3-There many i.g. and e.g. during the article. After a while, They somehow bother the reader. I suggest you diminish the number of them.

4- On the 87. line, the correction is needed: "the families from LOW socioeconomic status.... 

5- The sentence between 122-124 lines (starting with "For example...")  is not completed. I could not understand what do you mean in that sentence.

6- Sometimes you wrote "at distance" but sometimes "at-distance". It should be in one form.

7-One of the most important deficiency is that I could not understand why this research matters to people. I can see that you wrote, "this study aims to analyze fluctuations...". Nevertheless, which gap in the field you will fulfill? It is not clear to me.

8- I think the effects of Covid-19 closures on reading and learning should be included more than it is now in the literature review. Although you cited many studies about school closures, they were mostly closures other than the Covid-19 process. What does other research find? What the effects will be for the students in the third grade?

9-It is really hard to understand your research questions (lines between 143-152) because there are some confusing aspects. Firstly, there is a lack of punctuation I think. Secondly, you used question marks but the sentences are not in question form. Thirdly, it is not clear which M1 or M2 belongs to which sentences. 

10-The method section is the most confusing part. As a general acceptance, the method of your research is better given at the beginning of the section. I looked for the method, however, it was at the last title. 

11-What is the difference between the northern and southern schools? I mean, are they different according to SES? How many students in the study have ASE and how many of them do not? What is the criterion to have ASE? As a non-Portugal academic, I do not have a clear understanding of ASE and students with ASE. Because one of the main purposes was to compare students based on their ASE situation, readers need more explanation.

12-The data collection procedures are not clear enough in my opinion. In the second paragraph of "Data collection and procedures" title, I am not sure whether you explained data collection from students, teachers, or both. Separating the paragraph depending on the groups may make it more clear.

13-Under the title numbered 2.2.1, there is much lack of information. These are listed below:

-Which reading errors did you take into account when you measure accuracy and speed?

-You said you calculated Flesh Reading Ease Formula. What was the result? What does that formula mean?

-Why teachers measured the prosody? Did they have any instructions about assessing it? If yes, how and by who? If not, how can we trust the results? This is a real concern. In the field of reading fluency, many articles provide information about instruction that assessors of prosody take. Teachers without required instruction may score biased. I think this bias may explain why the students were found to have a lower level of speed but a higher level of prosody. Because the assessment of prosody depends on more subjective criteria, the teachers may give more scores to their students. The students were at normative scores in accuracy, were lower in speed, and were higher in prosody. How do you explain this? It is an unexpected situation I guess as students are expected to read accurately, and then fast,  and lastly prosodically (see Fluency Instruction book). Is there any explanation for this contradiction?

-You mentioned that you collected the readings of students from Zoom, Skype, etc. Did students hear each other when one of them read the text? How did you prevent them to listen to each other?

-You did not provide interrater reliability for prosody assessments. Why?

-It is better for readers to know more about the "Reading Academic Target Skills" procedure. You mentioned that teachers used a checklist. What kind of checklist was that? What does it include? How it can be used? You are familiar with it, however, I can not understand what it includes, who developed it, etc. Secondly, what is the interrater reliability of these assessments?

14-In table 1, there are two many "With ASE support". One must be "Without ASE support".

15- What does DV's mean? (line 319)

16- In the findings, you said students were below the speed when compared to normative scores. However, we do not know your normative scores. It is important for us readers to know your criteria for reading fluency.

17-What does DifM means? (line 323) 

18- In the discussion part, there may be some misconclusions. They are listed below:

-(Line 482) You wrote the reading fluency scores of students were lower than expected. However, findings showed that they were lower only in speed. How can you conclude that they were lower in reading fluency as a whole?

-(Line 484) Similarly, the ii conclusion is not correct if I consider your findings.

I think the discussion part should be improved. and checked. There are nine resources in the discussion part. This might lead to relatively restricted discussion, I think. This article, in my opinion, deserves a more comprehensive discussion.

I am very happy to read this article.

Best Regards.

Author Response

Manuscript children-2001652

Response to Reviewer 1

Dear Reviewer,

Thank you for allowing us the possibility to submit a revised draft of the manuscript “Reading in COVID-19 pandemic times: a snapshot of reading fluency of Portuguese elementary school students” for publication in Children. We appreciate the time and effort the reviewers have dedicated to reading the manuscript and providing us with feedback. We are grateful for the insightful comments and valuable insights that helped improve our paper. We made our best efforts to incorporate the suggestions made by the reviewers. Those changes are highlighted in the manuscript. Please see below, in blue, a point-by-point response to the reviewer’s comments and concerns. All page numbers refer to the revised manuscript file with changes highlighted in red.

Reviewer’s comments to the authors:

Reviewer 1:

Dear Authors,

I am very pleased to review such an article. I think it touches on a crucial title.

However, as a non-English speaker, I have many confusions and questions in my mind. In my opinion, some parts of the article should be improved.

  1. In the abstract, you said 22 third graders included. In the methods section, you wrote 52. 

Thank you very much for bringing this information to our attention. We apologize for our mistake; in the abstract, we mixed the average number of students per class with the number of participants. The number of participants is 52 (26 male and 26 female). This was changed accordingly in the manuscript.

  1. In the abstract, you mentioned accuracy and speed but skipped prosody and the other two variables. Why?

Thank you for this observation. We recognize that we didn’t mention all variables in the abstract. We have already added those results in the abstract.

  1. There many i.g. and e.g. during the article. After a while, They somehow bother the reader. I suggest you diminish the number of them.

Thank you for this observation. We have already decreased throughout the manuscript the number of “i.e.,” and “e.g.,”.

  1. On the 87. line, the correction is needed: "the families from LOW socioeconomic status.... 

Thank you for the warning. This word was included in the sentence.

  1. The sentence between 122-124 lines (starting with "For example...")  is not completed. I could not understand what do you mean in that sentence.

This sentence was rewritten for clarity purposes.

  1. Sometimes you wrote "at distance" but sometimes "at-distance". It should be in one form.

Thank you for this observation. The new version uses "at-distance” consistently.

  1. One of the most important deficiency is that I could not understand why this research matters to people. I can see that you wrote, "this study aims to analyze fluctuations...". Nevertheless, which gap in the field you will fulfill? It is not clear to me.

Thank you for pointing this out. We agree that the argument explaining how this research was helping to fill in the gap in the field needed further clarification. We added a sentence to clarify our point of view.

  1. I think the effects of Covid-19 closures on reading and learning should be included more than it is now in the literature review. Although you cited many studies about school closures, they were mostly closures other than the Covid-19 process. What does other research find? What the effects will be for the students in the third grade?

Thanks for this comment; we agree that we have not focused on the COVID-19 process as readers would expect. We changed the text accordingly to better explain the effects of the pandemic, especially about the 3rd-graders’ educational needs.

  1. It is really hard to understand your research questions (lines between 143-152) because there are some confusing aspects. Firstly, there is a lack of punctuation I think. Secondly, you used question marks but the sentences are not in question form. Thirdly, it is not clear which M1 or M2 belongs to which sentences. 

Thank you for the feedback. The paragraph was rewritten for clarity reasons.

  1. The method section is the most confusing part. As a general acceptance, the method of your research is better given at the beginning of the section. I looked for the method, however, it was at the last title.

 Thank you for pointing this out. We agree with your observation and rearranged the method section in order to be more apparent.

  1. What is the difference between the northern and southern schools? I mean, are they different according to SES? How many students in the study have ASE and how many of them do not? What is the criterion to have ASE? As a non-Portugal academic, I do not have a clear understanding of ASE and students with ASE. Because one of the main purposes was to compare students based on their ASE situation, readers need more explanation.

We thank the reviewer for this comment. We added information to clarify these questions and help non-Portuguese readers.

  1. The data collection procedures are not clear enough in my opinion. In the second paragraph of "Data collection and procedures" title, I am not sure whether you explained data collection from students, teachers, or both. Separating the paragraph depending on the groups may make it clearer.

Thank you. We have clarified the paragraph to make it easier to understand.

  1. Under the title numbered 2.2.1, there is much lack of information. These are listed below:
    • Which reading errors did you take into account when you measure accuracy and speed?

Thank you for this comment and suggestion made. We have clarified the type of errors considered, which was very useful for explaining the methodology.

  • You said you calculated Flesh Reading Ease Formula. What was the result? What does that formula mean?

We thank the reviewer for this question. We clarified the results and the formula meaning in the manuscript.

  • Why teachers measured the prosody? Did they have any instructions about assessing it? If yes, how and by who? If not, how can we trust the results? This is a real concern. In the field of reading fluency, many articles provide information about instruction that assessors of prosody take. Teachers without required instruction may score biased. I think this bias may explain why the students were found to have a lower level of speed but a higher level of prosody. Because the assessment of prosody depends on more subjective criteria, the teachers may give more scores to their students. The students were at normative scores in accuracy, were lower in speed, and were higher in prosody. How do you explain this? It is an unexpected situation I guess as students are expected to read accurately, and then fast, and lastly prosodically (see Fluency Instruction book). Is there any explanation for this contradiction?

We thank this reviewer’s comment. We agree that there is a need to explain this result further. Moreover, we agree that teachers may have benefited their students by being generous with their prosody scores. We added this reflection to the discussion section.

  • You mentioned that you collected the readings of students from Zoom, Skype, etc. Did students hear each other when one of them read the text? How did you prevent them to listen to each other?

Thank you for your observation. Data were collected when the students were at home and on an individual basis; therefore, students did not listen to each other. In the case of the second moment, even though they were already at school, the collection was individual and through an online mode.

  • You did not provide interrater reliability for prosody assessments. Why?

We thank the reviewer for this question. We added these data to the new version of the manuscript.

  • It is better for readers to know more about the "Reading Academic Target Skills" procedure. You mentioned that teachers used a checklist. What kind of checklist was that? What does it include? How it can be used? You are familiar with it, however, I can not understand what it includes, who developed it, etc. Secondly, what is the interrater reliability of these assessments?

Thank you for the comment. We appreciate the opportunity to explain this topic further. We have changed and added information in the paragraph to help readers better understand the procedures followed.

  1. In table 1, there are two many "With ASE support". One must be "Without ASE support".

Thank you for the correction. The expression was corrected.

  1. What does DV's mean? (line 319)

Thank you. We have clarified the abbreviation.

  1. In the findings, you said students were below the speed when compared to normative scores. However, we do not know your normative scores. It is important for us readers to know your criteria for reading fluency.

We would like to thank the reviewer for the comment and the opportunity to clarify this aspect of the paper. In the 3rd grade students are expected to read about 110 words per minute. We have added this information in the Quantitative results section.

  1. What does DifM means? (line 323) 

Thank you. We have clarified the abbreviation.

  1. In the discussion part, there may be some misconclusions. They are listed below:
    • (Line 482) You wrote the reading fluency scores of students were lower than expected. However, findings showed that they were lower only in speed. How can you conclude that they were lower in reading fluency as a whole?

We thank the reviewer for this comment. We have clarified the sentences for better understanding.

  • (Line 484) Similarly, the ii conclusion is not correct if I consider your findings.

We thank the reviewer for this comment. This sentence was also clarified to ensure a better understanding.

I think the discussion part should be improved. and checked. There are nine resources in the discussion part. This might lead to relatively restricted discussion, I think. This article, in my opinion, deserves a more comprehensive discussion.

We thank the reviewer for this comment. The discussion section was revisited and the text was improved to convey a clear message.

Reviewer 2 Report

The article presents a very interesting and relevant study on how a prolonged period of absence from school affects the reading fluency of Portuguese primary school pupils and whether the effects differ depending on whether the pupils come from families with state financial support or not. The attempt to achieve deeper insights into the results and background of the findings through a mixture of quantitative and qualitative research is commendable.

The introductory remarks summarise the state of the research well and illustrate the relevance of the research subject well. The selection of participants and the conduct of the study should be described in more detail at certain points (see further comments). The references are appropriate, comprehensive, refer to relevant sources and are sufficiently up-to-date. There are no excessive self-citations. The statements and conclusions drawn are coherent and supported by the listed citations.

Novelty: The question is original and well-defined. The results provide an advancement of the current knowledge

Scope: The work fits the journal's scope.

Significance: The results seem to be interpreted appropriately and they are significant. All conclusions seem justified and supported by the results. The hypotheses seem to be carefully identified as such.

Quality: The article is written in an appropriate way and the data and analyses are presented appropriately.

Scientific Soundness: The study seems mostly correctly designed and technically sound (please refer to the questions mentioned further below). The methods, tools, software, and reagents are described with sufficient detail to allow another researcher to reproduce the results.

Interest to the Readers: The conclusions are very interesting for the readership of the journal. Supposedly, the paper will attract quite a wide readership.

Overall Merit: there is definitely an overall benefit to publishing this work since it advances the current knowledge.

English Level: The English language is mostly appropriate and understandable (see further explanations below).

Regarding the presentation of the study, I have the following questions to be considered in the revision of the article:

Participants: The abstract mentions 22 third graders (l. 25). Later, however, 52 participants are mentioned (L. 165). Regarding the participating schools and classes, it would be interesting and important to know according to which criteria they were selected, e.g. were they schools known to the authors; if there were several third-grade classes at one school, which one was selected and why? How were the pupils randomly selected? How were the teachers assigned to the two groups of schools and thus to the focus groups; what criteria, if any, were decisive for this?

Dates of the survey: In line 143 June is given as M1, in line 195 July, please clarify the contradiction or standardise dates.

Data collection:

Regarding reading fluency: it is not clear who collected the reading fluency. Was it the teachers and the two researchers (L. 240-243) or only the researchers? And was interrater reliability only measured between the researchers? What agreement, if any, was there with the teachers' assessment?

Why were three texts used at only two data collection points? How were the three (?) texts assigned to the classes?

Regarding prosody: How was it ensured that the teachers involved could assess the prosody as similarly as possible? Were they trained to use the instrument before the start of the study (l. 219-223)?

Regarding the Reading academic target skills: here too, the question of training the study participants, i.e. the teachers, arises. This seems particularly relevant as later this is the area where statistically significant differences in perceived quality were found (L. 374-376).

Presentation of the results: Regarding Table 1: It says "With ASE Support" twice, shouldn't it say "Without ASE Support" once?

Stylistically, the formulation of the research questions as statements with question marks at the end of the sentence is surprising (L. 144-151). Wouldn't it be more appropriate to formulate them as questions, especially since this is also done for the last question (L. 151-152)?

In terms of language and style, the text is largely good. Occasionally words (e.g. l. 242) or punctuation marks (l. 361) are missing.

Author Response

Reviewer 2:

The article presents a very interesting and relevant study on how a prolonged period of absence from school affects the reading fluency of Portuguese primary school pupils and whether the effects differ depending on whether the pupils come from families with state financial support or not. The attempt to achieve deeper insights into the results and background of the findings through a mixture of quantitative and qualitative research is commendable.

The introductory remarks summarise the state of the research well and illustrate the relevance of the research subject well. The selection of participants and the conduct of the study should be described in more detail at certain points (see further comments). The references are appropriate, comprehensive, refer to relevant sources and are sufficiently up-to-date. There are no excessive self-citations. The statements and conclusions drawn are coherent and supported by the listed citations.

Novelty: The question is original and well-defined. The results provide an advancement of the current knowledge

Scope: The work fits the journal's scope.

Significance: The results seem to be interpreted appropriately and they are significant. All conclusions seem justified and supported by the results. The hypotheses seem to be carefully identified as such.

Quality: The article is written in an appropriate way and the data and analyses are presented appropriately.

Scientific Soundness: The study seems mostly correctly designed and technically sound (please refer to the questions mentioned further below). The methods, tools, software, and reagents are described with sufficient detail to allow another researcher to reproduce the results.

Interest to the Readers: The conclusions are very interesting for the readership of the journal. Supposedly, the paper will attract quite a wide readership.

Overall Merit: there is definitely an overall benefit to publishing this work since it advances the current knowledge.

English Level: The English language is mostly appropriate and understandable (see further explanations below).

Regarding the presentation of the study, I have the following questions to be considered in the revision of the article:

               Participants: The abstract mentions 22 third graders (l. 25). Later, however, 52 participants are mentioned (L. 165). Regarding the participating schools and classes, it would be interesting and important to know according to which criteria they were selected, e.g. were they schools known to the authors; if there were several third-grade classes at one school, which one was selected and why? How were the pupils randomly selected? How were the teachers assigned to the two groups of schools and thus to the focus groups; what criteria, if any, were decisive for this?

       Thank you very much for bringing this information to our attention. We apologize for our mistake; in the abstract, we mixed the number of participants with the average number of students per class. The number of participants is 52 (26 male and 26 female).

The schools chosen were familiar to the authors because they are connected to other projects developed by the research team. The process followed was the easiest way to reach the participants. The classes selected had already been enrolled in previous educational projects run by the authors but focused on a different topic. Within each class, we randomized students using their school numbers and online randomization software. The teachers selected were the titular teachers of these same classes and, automatically, were those who participated in the focus groups.

Dates of the survey: In line 143 June is given as M1, in line 195 July, please clarify the contradiction or standardise dates.

Thank you very much for bringing this information to our attention and for the opportunity to correct this aspect in the paper. In fact, the correct date is June. We have already changed the information accordingly.

Data collection:

Regarding reading fluency: it is not clear who collected the reading fluency. Was it the teachers and the two researchers (L. 240-243) or only the researchers? And was interrater reliability only measured between the researchers? What agreement, if any, was there with the teachers' assessment?

               Thank you for this observation. We recognize that this information could have been delivered better. Students were assessed by their class teacher. We have already added this information in the new version of the manuscript.

Why were three texts used at only two data collection points? How were the three (?) texts assigned to the classes?

Thank you for the question. This sentence was clarified.

Regarding prosody: How was it ensured that the teachers involved could assess the prosody as similarly as possible? Were they trained to use the instrument before the start of the study (l. 219-223)?

               Thank you for the question. Actually, the scale used by the teachers provided a very detailed explanation of the criteria for every point in the scale. Moreover, Portuguese teachers are familiar with these criteria because they use them regularly in their educational practice.

Regarding the Reading academic target skills: here too, the question of training the study participants, i.e. the teachers, arises. This seems particularly relevant as later this is the area where statistically significant differences in perceived quality were found (L. 374-376).

We thank the reviewer for this comment. This part of the study was changed to convey a clear message and help readers understand the procedures followed.

Presentation of the results: Regarding Table 1: It says "With ASE Support" twice, shouldn't it say "Without ASE Support" once?

       Thank you for the correction. The text was changed accordingly.

Stylistically, the formulation of the research questions as statements with question marks at the end of the sentence is surprising (L. 144-151). Wouldn't it be more appropriate to formulate them as questions, especially since this is also done for the last question (L. 151-152)?

Thank you for this observation. We recognize that the research questions need to be clearer. We re-written all questions in order to be more comprehensive.

In terms of language and style, the text is largely good. Occasionally words (e.g. l. 242) or punctuation marks (l. 361) are missing.

Thank you for the observation. The sentence was rewritten.

Finally, the manuscript was edited by a native speaker.